# Effects of Probiotic Supplementation on Body Weight, Growth Performance, Immune Function, Intestinal Microbiota and Metabolites in Fallow Deer

**DOI:** 10.3390/biology13080603

**Published:** 2024-08-09

**Authors:** Meihui Wang, Qingyun Guo, Yunfang Shan, Zhibin Cheng, Qingxun Zhang, Jiade Bai, Yulan Dong, Zhenyu Zhong

**Affiliations:** 1National Key Laboratory of Veterinary Public Health Security, College of Veterinary Medicine, China Agricultural University, Beijing 100193, China; wangmeihui718@163.com (M.W.); yulandong77@163.com (Y.D.); 2Milu Conservation Research Unit, Beijing Milu Ecological Research Center, Beijing 100076, China; shanyunfang@yeah.net (Y.S.); miluchengzb@163.com (Z.C.); baijiade234@aliyun.com (J.B.)

**Keywords:** *Enterococcus faecalis*, probiotic, deer, intestinal microbiota, metabolomics analysis

## Abstract

**Simple Summary:**

China’s deer population is rich in resources and species; how to improve the health status of the deer population and reduce the occurrence of intestinal diseases is a necessary direction for current research. In this study, *Enterococcus faecalis* was used as the research object to explore the effect of lactic acid bacteria on deer growth performance and intestinal health and to further investigate the role played by lactic acid bacteria on the intestinal health status of deer. The results of this study showed that the addition of *Enterococcus faecalis* to the feed promoted weight gain, improved immune function, altered the structure of the intestinal flora of fallow deer, especially increasing the abundance of *Firmicutes* phylum, and increased the abundance of the *Ruminococcaceae* and *Lacertococcaceae* at the same phylum as that of *Enterococcus faecalis*. Furthermore, metabolomic analyses also revealed that *Enterococcus faecalis* supplementation increased the metabolism of lipids, carbohydrates and phospholipids by the intestinal microbiota. In conclusion, this study revealed the beneficial effects of lactic acid bacteria on growth traits, immunity and intestinal health of deer and provided a scientific basis for the further development of deer-specific probiotic products to promote intestinal health and reduce intestinal diseases in deer species in the future.

**Abstract:**

Intestinal diseases are one of the diseases that affect the growth and immunity of deer. Currently, more lactic acid bacteria (LAB) are available as feed additives to improve the intestinal ecological balance of ruminants in production practices. In this study, *Enterococcus faecalis* was supplemented in the feed of fallow deer for 170 d, and body weights, blood indices and immune levels of fallow deer were counted at 35, 65 and 170 d. The effects of *Enterococcus faecalis* on the intestinal microbiota and the metabolism of fallow deer were analysed using 16S rDNA and UPLC-MS/MS methods. The results showed that the addition of *Enterococcus faecalis* to the diet improved body weight and immune function and increased the aggregation of gut microbiota in fallow deer. The addition of *Enterococcus faecalis* altered the community structure of intestinal microorganisms in fallow deer and increased the number of beneficial bacteria. In addition, combined with metabolomics analysis, it was found that supplementation with *Enterococcus faecalis* significantly altered the metabolites of fallow deer, mainly regulating lipid metabolism, carbohydrate metabolism and phospholipid metabolism. In conclusion, this study presents, for the first time, evidence that the LAB strain *Enterococcus faecalis* can be used as a potential probiotic for deer and points to a new direction for the treatment of intestinal disorders in the deer family.

## 1. Introduction

Deer breeding has a long history in China, and the breeding of deer is not only closely related to material and cultural needs but is also an extension of the traditional Chinese medicine health industry. According to 2021 statistics, the total economic value of deer breeding and its upstream and downstream industries is around 35 billion yuan per year, which is an important part of China’s animal husbandry industry [1]. After China’s Ministry of Agriculture and Rural Development (MARD) announced in May 2020 that sika deer (*Cervus nippon*) and red deer (*Cervus canadensis*) were included in the management of MARD as special livestock and poultry, the breeding scale of sika deer and red deer was further expanded [2]. It is worth noting that as a national-level protected animal, the conservation and population recovery of Milu (*Elaphurus davidianus*) has been of great concern in China [3,4]. In recent years, the frequent occurrence of deer bacterial intestinal diseases, such as *Clostridium perfringens* [5], pathogenic *Escherichia coli* [6], *Bacillus cereus* [7], *Salmonella* [8] and *Pasteurella* [9], which usually co-infect deer, poses a serious threat to the survival of deer. For the past several decades, bacterial disease control has been conducted through the use of chemicals, primarily antibiotics. Nevertheless, antibiotic residues are recognised as a serious threat due to their persistence in the animal body and the increase in resistance [10,11]. These problems, including alteration of the gut microbiota, disruption of the living environment and the emergence of drug-resistant bacterial populations, have been reported to be associated with the misuse of antibiotics, with unpredictable long-term consequences for public health [12]. Alternatives to antibiotics are urgently needed to prevent and/or intervene in deer bacterial disease outbreaks. As a result, probiotics are now considered a reliable prospective option [13].

Earlier, the World Health Organization (WHO) and the Food and Agriculture Organization (FAO) of the United Nations jointly advocated the use of probiotics as live microorganisms or cultured product feed supplements and, when consumed in moderation, can have a beneficial effect on the health of animals [14]. Probiotics are commonly known to be advantageous in increasing body weight, enhancing the immune response, providing nutrient and enzyme contributions to the host, and balancing the gut microbiota [15]. Probiotics have a wide range of applications in ruminants, such as dairy cows, deer and lambs [16]. Among the many probiotics that have been used as feed supplements, lactic acid bacteria (LAB) have been extensively studied in recent years. LAB are a group of bacteria that produce large amounts of lactic acid from fermentable carbohydrates, the vast majority of which are essential and physiologically important organisms in the body, and are widely found in the intestine of humans and animals [17]. A number of LAB strains are used as probiotics in deer farming with the aim of enhancing immunity and disease resistance, modulating the gut microbiota and competitively excluding pathogens through the production of inhibitory substances, such as *Lactobacillus plantarum* [18], *Lactobacillus* SCG 1223 [19], *Lactobacillus casei* Zhang [20], etc.

*Enterococcus faecalis* is a parthenogenetic anaerobic gram-positive LAB and an inherent species of bacteria in the gastrointestinal tract of humans and animals [21]. The Catalogue of Feed Additive Species (2013) specifies *Enterococcus faecalis* as a strain that can be added to feed [22]. *Enterococcus faecalis* has the efficacy of protecting the intestinal barrier, which forms biofilm on the intestinal epithelial cells and protects them from damage caused by harmful substances and plays a probiotic role [23]. At the same time, *Enterococcus faecalis* has nutritional efficacy, and studies have shown that the bacterium contains a variety of small peptides that facilitate the absorption of amino acids and residues and improves the ability of tissues to synthesise proteins [24,25]. In addition, the bacterium can break down some proteins into amides and amino acids, and it can soften the fibres in feeds and improve the conversion and absorption rate of feeds [26]. As a result, *Enterococcus faecalis* is now also widely used as a feed additive on farms as a way to regulate the animal’s gut microbiota, improve feed efficiency and growth rates and reduce the use of antimicrobials [27,28]. Little is known about the use of *Enterococcus faecalis* as a LAB in deer diets, although a great deal of work has been conducted to investigate the role of *Enterococcus faecalis* in intestinal growth performance, feed nutrient utilization, disease resistance and the environment of the gut microbiota in ruminants.

The purpose of this study was to evaluate the potential role of LAB (*Enterococcus faecalis*) in deer growth performance and disease resistance. The results of this study indicate that adding *Enterococcus faecalis* to feed improved deer growth rates and immune responses, and the results of these experiments could provide essential information about the use of probiotics in protected animals (such as Milu) and deer husbandry.

## 2. Materials and Methods

### 2.1. Animal Management

Twelve 2-month-old fallow deer were randomly divided into 2 groups. The control group (*n* = 6) received only a basic diet, and the treatment group (*n* = 6) received a basic diet for the entire experimental period (170 days) and *Enterococcus faecalis* (Okobaike Biotechnology Co. Ltd., Luoyang, China) (≥2.0 × 10^10^ CFU/g) in a mixed diet. Specifically speaking, the treatment group had a mixed diet of 22% concentrate (Junwei, Tianjin, China), 26% carrots, 27% dry alfalfa grass, 25% silage and 0.002% *Enterococcus faecalis* (≥2.0 × 10^8^ CFU/g). The deer concentrate feed comprised a feed formulation of 50% maize, 26% soya bean meal, 11% bran, 10% barley, 2% calcium phosphate and 1% salt. This mixed diet covers the recommended nutritional requirements [29,30]. Each deer consumes approximately 500 g of feed at the first feeding instance and then feeds itself. The diet was then adjusted daily based on the previous day’s leftovers to ensure that approximately 1% of the feed was left in the trough at the second day feeding. Throughout the experimental period, fallow deer were fed twice a day at 9:00 and 15:00. Furthermore, 12 fallow deer were placed in the same pen throughout the 170-day experiment. The experiment was carried out under proper lighting and ventilation conditions and with appropriate management. All fallow deer were not vaccinated and were not infected with other bacteria. All procedures of this 170-day experiment were carried out with approval by the Ethics Review Committee of Experimental Animal Welfare and Animal Experimentation of the China Agricultural University (approval no. CAU201709112).

### 2.2. Weight Determination

At 0 d, fallow deer were weighed 2 h before morning feeding as the initial weight. At 35 d, 65 d and 170 d, all animals were weighed 2 h before morning feeding, and growth curves for fallow deer were constructed using days as the horizontal axis and body weight as the vertical axis. Gain weight was obtained by subtracting the initial weight from the final body weight. Daily weight gain is the gain weight divided by the number of days in the study.

### 2.3. Faecal Collection and Handling

Fallow deer faeces were collected according to the sample collection procedure of Zhang et al. [31]. Fresh faecal samples were collected in the morning at 35 d, 65 d and 170 d. Samples were collected with sterile gloves and were stored in sealed sterile centrifuge tubes to avoid cross contamination between samples. After sampling, samples were frozen in liquid nitrogen and then preserved at −80 °C.

### 2.4. Blood Collection and Processing

Blood samples from all fallow deer were collected from each group on the morning of day 35, 65 and 170. Blood samples were collected and placed in collection tubes containing anticoagulant; the anticoagulant tubes were gently shaken upside down. After that, we measured the plasma levels of blood with a BC-2800Vet Haematology Analyzer (Mindray, Shenzhen, China). These included white blood cells (WBC), lymphocytes (Lymph), monocytes (Mon), neutrophils (Gran), red blood cells (RBC), haemoglobin (HGB), red blood cell pressure (HCT), platelet count (PLT) and platelet corpuscle (PCT).

Serum was prepared by taking approximately 5 mL of blood into a collection tube without anticoagulant which was then left to clot naturally for 30–60 min at room temperature. After the blood coagulated, it was centrifuged in a centrifuge (Eppendorf, Hamburg, Germany) at 3000 rpm for 10 min, and the supernatant was aspirated as serum. Serum is analysed biochemically using biochemical marker kits (Zhongsheng Beizhong Biotechnology Co., Ltd., Beijing, China) including total protein (TP), albumin (ALB), globulin (GLB), albumin/globulin (A/G), total cholesterol (TC), triglycerides (TG), aspartate aminotransferase (AST), alanine aminotransferase (ALT), glucose (GLU), lactate dehydrogenase (LDH), serum calcium and serum phosphorus. Furthermore, a deer Immunoglobulin IgG/A/M Assay Kit (Meimian, Yancheng, China) was used. Immunoglobulin (IgA, IgM, IgG) levels were measured. The ranges for the deer Immunoglobulin IgG/A/M Assay Kit were IgG 8.53 ± 3.197 g/L, IgA 1.248 ± 0.54 g/L and IgM 0.911 ± 0.302 g/L. The other portion of serum was stored in −20 °C for subsequent experiments.

### 2.5. Enzyme-Linked Immunosorbent Assay (ELISA)

ELISA kits (Cloud-Clone Corp., Wuhan, China) were used to detect the levels of the immunological factors TNF-α, IL-4, IL-6 and IL-8 in serum, and the specific steps, measurements and calculations were performed according to the operation manual. The substances detected in the TNF-α, IL-4, IL-6 and IL-8 kits showed no significant cross-reactivity with other similar substances, and the CV within the assay was <10%. The ranges of the ELISA kits were IL-4 15.6–1000 pg/mL, IL-6 7.8–500 pg/mL, IL-8 15.6–1000 pg/mL and TNF-α 7.8–500 pg/mL.

### 2.6. Faecal DNA Extraction, PCR Amplification and Sequencing of Intestinal Microbiota

Total faecal DNA was extracted with the TlANamp Stool DNA Kit (Tiangen Biotech, Beijing, China) according to the instructions of the manufacturer, purified and tested for DNA concentration and purity via Nanodrop (Thermo Fisher Scientific, Waltham, MA, USA). The extracted DNA samples were used as templates for PCR amplification of the bacterial 16S rDNA V3–V4 region to construct libraries for PacBio sequencing. Reads were then spliced and filtered, clustered or denoised, as well as subjected to species annotation and abundance analyses to reveal the species composition of the samples. The preparation and sequencing of 16S rDNA of faecal microbiota were performed on the Biomarker Biotechnology platform (https://www.biocloud.net/) (accessed on 18 June 2024), such as α-diversity, β-diversity, microbial aggregation analysis, correlation analysis and community abundance analysis. Using the Usearch software (version 11.0) [32], Clustering of Reads at 97.0% similarity level was conducted, obtaining OTUs. using the SILVA database (v.123) the classifications were assigned to OTUs with a confidence level of 70% for the RDP classifiers [33]. A-diversity indices including ACE, Chao1, Shannon and Simpson indices were calculated using QIIME2 software (v.2020.8). Statistical significance between groups was determined using one-way analysis of variance (ANOVA) (version 20.0). The structure of microbial communities among samples was analysed with β-diversity analysis using the binary_jaccard distance metric and visualized with principal component analysis (PCA), principal coordinate analysis (PCoA), non-metric multidimensional scaling (NMDS) and the unweighted pairwise grouping method (UPGMA).

### 2.7. Non-Targeted Metabolomics Analysis of Intestinal Microbiota Metabolites

The ultra-performance liquid chromatograph–mass spectrometer (UPLC–MS/MS) system for metabolomics analysis of faecal samples consisted of an Acquity I-Class PLUS ultra-performance liquid chromatographer (Waters Technology Co., Milford, MA, USA) coupled with a Xevo G2-XS QTOF high-resolution mass spectrometer (Waters Technology Co., Milford, MA, USA), using a Waters Acquity UPLC HSS T3 column. Firstly, the metabolites were extracted into the extract with an acetonitrile–water volume ratio of 1:1, and the samples were centrifuged to mix the samples into QC samples for on-line assay. Then, the on-line assay was performed in positive and negative ion modes with a capillary voltage of 2500 V or −2000 V, a temperature of 100 °C, a flow rate of 400 μL/min and an injection volume of 1 μL, respectively. The Waters Xevo G2-XS QTof high-resolution mass spectrometer was used for primary and secondary mass spectrometry data acquisition in MSe mode under Waters control via MassLynx V4.2 acquisition software [34]. Raw data collected via MassLynx V4.2 were processed with Progenesis QI 2.0 software for peak extraction and peak alignment and then identified against the METLIN database and Bemac’s own library. The data were normalised before analysis via total peak area normalisation, i.e., each metabolite in each sample was divided by the total peak area of the sample and multiplied by the mean of all peak areas [35]. Metabolomics analyses were analysed using the bioinformatics analysis process of the BMKCloud platform (https://www.biocloud.net/) (accessed on 18 June 2024).

### 2.8. Statistical Analyses

In this experiment, all experimental results were analysed using SPSS 20.0 and presented in the form of mean ± SEM. Figures were generated with GraphPad Prism 8.1 (San Diego, CA, USA) and R Studio (version 4.0.3). Experiments were performed in at least five independent biological replicates. Prior to data analysis, the test data were all normally distributed according to the Shapiro–Wilk test. For the detection of weight, blood parameters and immunoglobulins, simple main effects and interaction effects between time and group were tested using a 3 (time: 35 d, 65 d, 170 d) × 2 (group: treatment and control) mixed factorial ANOVA. For the detection of IL-4, IL-6, IL-8 and TNF-α, the difference between the experimental values of the two groups was determined using the independent *t*-test. Statistical significance was as follows: * *p* < 0.05, ** *p* < 0.01 and *** *p* < 0.001.

## 3. Results

### 3.1. Effect of Enterococcus faecalis Supplementation on Body Weight Parameters of Fallow Deer

Body weight, total weight gain and average daily weight gain directly reflect the growth performance of animals. We measured the body weights of fallow deer in the control group and treatment group at 0 d, 35 d, 65 d and 170 d, respectively (Figure 1). After 35 d of consumption of feed containing *Enterococcus faecalis* by 2-month-old weaned fallow deer, the weight of fallow deer in the treatment group significantly increased by 36.46% (*p* < 0.001); the total weight gain was significantly higher than that of fallow deer in the control group by 733.96% (*p* < 0.001), and the average daily gain was significantly higher than that of fallow deer in the control group by 550% (*p* < 0.001). Similarly, after 65 d of feeding, body weight, total weight gain and average daily weight gain of fallow deer in the treated group were significantly higher by 23.07% (*p* < 0.001), 59.97% (*p* < 0.001) and 55.56% (*p* < 0.001), respectively, as compared to the control group. After 170 d of feeding, the body weight of fallow deer in the treatment group was elevated by 10.14% (*p* < 0.05), total weight gain was significantly higher by 14.48% (*p* < 0.05), and average daily weight gain was significantly higher by 13.64% (*p* < 0.05). As shown in the body weight growth curve, the treatment group grew rapidly from 0 d to 65 d, while the growth rate slowed down from 65 d to 170 d (Figure 1B). This suggests that the addition of *Enterococcus faecalis* to the diet can increase the body weight of fallow deer, especially for young fallow deer, which in turn promotes the growth and development of fallow deer.

### 3.2. Effect of Enterococcus faecalis Supplementation on Blood Indices in Fallow Deer

Blood routine and blood biochemistry were carried out in order to assess the changes in the number and morphological distribution of the fallow deers’ blood cells, as well as the various metabolites such as various ions, sugars, proteins and lipids in the blood. As can be seen from Table 1, after feeding the diet for 35 d, 65 d and 170 d, the WBC, Lymph, Mon and Gran numbers in the blood of fallow deer in the treatment group increased, especially after 65 d of feeding; the WBC and Lymph in the blood of fallow deer in the experimental group were markedly greater than those in the control group (*p* < 0.05). Meanwhile, RBC, HGB, PCV, PLT and PCT also increased in the blood of fallow deer after feeding *Enterococcus faecalis* for 35 d, 65 d and 170 d, with significant increases in PCV after 35 and 65 d of feeding (*p* < 0.05), and significant increases in PLT and PCT were observed after 170 d of feeding (*p* < 0.05).

In Table 2, we found that the addition of *Enterococcus faecalis* to the ration, starting from 35 d of feeding to day 170 of feeding, led to a noticeable increase in the serum levels of TP, ALB and GLB in fallow deer in the treatment group compared to the control group, with the TP level increasing from 1.16% to 13.32% (*p* > 0.05), the ALB level increasing from 3.12% to 13.69% (*p* > 0.05) and the GLB content increasing from 3.42% to 14.95% (*p* > 0.05) (only measures at day 170 were significant (*p* < 0.05)). However, compared with the control group, the A/G content was reduced, indicating that *Enterococcus faecalis* could improve the protein metabolism of the deer organism to a certain extent and enhance the immunity of weaned fallow deer.

In addition, ALT and AST are important indicators of liver function. In this experiment, *Enterococcus faecalis* had no significant effect on both ALT and AST activities, indicating that the feed had no adverse effect on liver function. In addition, serum TG and TC content can reflect the lipid metabolism of animals. LDH exists in all tissues and organs of the organism and plays a role in glycolysis, and its activity can reflect the status of glucose metabolism of the organism. The blood biochemical results showed that feeding *Enterococcus faecalis* had no significant effect on TC from 35 d to 170 d, but significantly increased TG content at 65 d (*p* < 0.01) and 170 d (*p* < 0.05), indicating that long-term feeding of this feed could promote fat digestion and metabolism; meanwhile, *Enterococcus faecalis* significantly decreased LDH activity at 65 d and increased blood glucose content (*p* < 0.01), suggesting that *Enterococcus faecalis* could improve the lipid metabolism and glucose metabolism of fallow deer. As shown in Table 2, *Enterococcus faecalis* significantly increased phosphorus levels (*p* < 0.05), suggesting that *Enterococcus faecalis* can promote deer bone growth.

### 3.3. Effect of Enterococcus faecalis Supplementation on Immunity Levels in Fallow Deer

Immune suppression is an important factor affecting the growth of newborn and weaned animals. Immunoglobulin is directly involved in the humoral immune response of the animal organism and is an important indicator of the immune status of the organism. Therefore, we examined the IgA, IgG and IgM contents in the serum of fallow deer in the treatment group and fallow deer in the control group, and the results are shown in Figure 2A–C. We found that the addition of *Enterococcus faecalis* to the ration partially increased the IgA and IgM levels in fallow deer plasma, significantly increasing the IgM (*p* < 0.001) level at 35 d, IgA (*p* < 0.05) and IgM (*p* < 0.05) levels at 65 d, and a significant increase in IgA content at 170 d (p < 0.05). Further, inflammatory factors in fallow deer serum were detected, including TNF-α, IL-4, IL-6 and IL-8. The results showed that IL-6 and IL-8 were notably increased in the serum of treated fallow deer compared to the control group at 170 d of feeding *Enterococcus faecalis*, suggesting that *Enterococcus faecalis* was able to improve the immune function and health of fallow deer.

### 3.4. Effects of Enterococcus faecalis Supplementation on the Intestinal Microbiota of Fallow Deer

To test whether the addition of *Enterococcus faecalis* to the feed regulated the intestinal microbiota, we collected fallow deer faeces for 16S rDNA gene sequencing in order to analyse the taxonomic composition of bacteria after the addition of *Enterococcus faecalis*. We obtained a total of 36 fallow deer faecal samples (*n* = 6) at 35 d, 65 d and 170 d, respectively, and sequencing generated V3-V4 16S rDNA gene profiles.

The α-diversity reflects the richness and diversity of the microbial community. There were no significant changes in Chao1, Shannon, ACE and Simpson indices in the remaining treatment group (*p* > 0.05, Figure 3A), except for a significant increase in the difference in Chao1 and ACE indices between the control group and the treatment group at 35 d (*p* < 0.05). The β-diversity of microbial communities was further evaluated using PCA, PCoA and NMDS to measure the degree of similarity. Figure 3B showed that at 35 d, the similarity of each sample in the control group was high and there was some similarity with the treatment group; while at 65 d and 170 d the distance between the samples of each group was separated, but the inter-group distance between the control group and the treatment group was far away, which indicated that there was a significant difference between the intestinal microbiota of the two groups.

Microbial aggregation analyses showed that microbial associations within the control group were tighter than the treatment group at 35 d of *Enterococcus faecalis* supplementation. Whereas at 65 d and 170 d of *Enterococcus faecalis* supplementation, microorganisms within the treatment group showed tighter associations, whereas the control group had sparse associations (Figure 4A). The results of UPGMA and relative abundance analyses of gut microbial communities showed higher gut microbiota richness at 35 d compared to 65 d and 170 d (Figure 4B,C). The results of taxonomic analyses of species at the phylum level showed that *Firmicutes* and *Bacteroidetes* were the dominant groups in the gut microbiota. At the genus level, *Ruminocoecaceae*_UCG-005, *Rikenellaceae*_RC9_gut_group, uncultured_bacterium_f_*Lachnospiraceae* and *Ruminococcaceae*_UCG-010 were the dominant groups (Figure 4C).

This showed that *Enterococcus faecalis* significantly contributed to the abundance of the *Firmicutes* phylum during fallow deer growth and increased the abundance of *Ruminocoecaceae* and *Lachnospiraceae* at the same phylum level as *Enterococcus faecalis*.

### 3.5. Effects of Enterococcus faecium Supplementation on Metabolites of the Intestinal Microbiota of Fallow Deer

Next, to investigate the effect of Enterococcus faecium supplementation on the metabolites of fallow deer intestinal microbiota, we performed metabolomic analyses of fallow deer faeces at 35 d and 170 d, respectively. At 35 d of feeding diets containing Enterococcus faecium, PCA and partial least squares discriminant analysis (PLS-DA) scatter plots showed significant clustering in faecal metabolite profiles between the control and treatment groups. The orthogonal partial least squares discriminant analysis (OPLS-DA) score table also showed that the two groups were significantly separated from each other (R2Y = 0.998 and Q2 = 0.882, Figure 5A). At 170 d of feeding diets containing *Enterococcus faecalis*, the PCA scatterplot showed significant clustering in faecal metabolite profiles between the control and treatment groups, but the two groups were closer to each other than they were at 35 d. The OPLS-DA score table also showed that the two groups were significantly separated from each other (R2Y = 0.997 and Q2 = 0.896, Figure 5B). Based on the statistical values, the volcano plots showed an upward and downward trend in the differential metabolites (*p* < 0.05, |log2FC| > 1) (Figure 5C). At 35 d, 37 metabolite expressions were up-regulated and 82 metabolite expressions were down-regulated in the treatment group compared to the control group, of which the most significantly elevated metabolites in the treatment group were 3-hydroxyoctadecanoyl-CoA and 1-Oleoyl-L-.alpha.-lysophosphatidic acid, and the most significantly reduced included Delta 8,14-Sterol, Dipicolinic acid and DG (16:1 (8Z)/18:18Z54). At 170 d, 134 metabolite expressions were up-regulated and 271 metabolite expressions were down-regulated in the treated group compared to the control group, with the most significantly elevated metabolites in the treated group being Iberin-N-acetyl-cysteine, 2-(alpha-D-Mannosyl)-3-phosphoglycerate and 2-Ethyl-5-imino-1-cyclopenten-1-ol, and the most significantly decreased metabolite was iso-A2E (11-cis).

In particular, we screened 50 metabolites for the greatest changes between the control and treated groups at 35 d and 170 d. Combined KEGG analyses showed that at 35 d, more differential metabolites were enriched for steroid biosynthesis, bile acid biosynthesis, phosphotransferase system, carbohydrate digestion and absorption, fatty acid metabolism, ascorbic acid and aldose metabolism, amino and nucleotide glucose metabolism, carbon fixation pathway in prokaryotes, insulin secretion and insulin resistance. At 170 d, more differential metabolites were enriched for the biosynthesis of alkaloids derived from histidine and purine, purine metabolism, caffeine metabolism, glyoxylate and dicarboxylate metabolism, folate biosynthesis, retinol metabolism and linoleic acid metabolism (Figure 5E).

## 4. Discussion

LAB is now widely used as the primary probiotic for prevention of gastrointestinal inflammation in ruminants due to its safety, reliability and species and functional diversity [36,37]. A previous study documented the important role of probiotics in the deer family, including control and inhibition of infectious pathogens, improvement of growth properties and activity of digestive enzymes, as well as enhancement of gastrointestinal resistance, cell surface hydrophobicity, and self-aggregation capacity [38]. The LAB strain *Enterococcus faecalis* is also considered a probiotic [39] and is widely used in feed. Nevertheless, studies related to the effects of *Enterococcus faecalis* on growth performance and immune responses in deer species are still limited. For this reason, in this study, we supplemented *Enterococcus faecalis* in fallow deer feed and collected fresh faeces from control and treatment groups at different times to reveal the probiotic effects of LAB in the diets of felids by evaluating body weights and immune-related factors and analysing the effects of *Enterococcus faecalis* on the intestinal metabolites of fallow deer via high-throughput 16S rDNA sequencing and metabolomics.

Numerous studies have shown that the addition of LAB to animal feeds can lead to weight gain [40]. This is due to the fact that probiotics can improve feed efficiency and help optimise the digestion and utilization of nutrients, promoting animals to extract more energy from the same amount of feed [41]. A study on the effect of *Lactobacillus* 6BZ and *Lactobacillus* 6BY on calf body weight showed that adding LAB to the diet increased the body weight of calves from birth to 5 months of age [42]. In addition, Jia et al. found that the addition of *Bacillus licheniformis* and *Saccharomyces cerevisiae* to fattened lambs significantly increased the average daily gain and had a beneficial effect on growth performance [43]. In this study, the addition of *Enterococcus faecalis* to the diet significantly increased body weight, total weight gain and average daily gain of fallow deer after 35 d of feeding compared to the control group (*p* < 0.001), indicating that the dietary addition of *Enterococcus faecalis* may improve the digestion of nutrients in fallow deer. Similarly, at 65 d, these indicators reflecting weight gain of fallow deer were significantly improved (*p* < 0.01). However, at 170 d, there was no significant difference between the body weights of fallow deer fed *Enterococcus faecalis* and the control group; the total weight gain and average daily weight gain were significantly higher (*p* < 0.05), but the increasing trend slowed down. The above results suggest that *Enterococcus faecalis* supplementation has a promoting effect on body weight gain and growth in young animals, and this promoting effect slows down as the age of the animals increases. Furthermore, the blood biochemical indexes of animals can reflect body metabolism, nutritional status and diseases of animals to a certain extent, thus indirectly reflecting the growth performance of animals. Serum TP consists of ALB and GLB, and its content can reflect the body’s protein metabolism. A high level of total protein indicates an increased ability of the liver to synthesise proteins, which has the effect of maintaining the osmotic pressure of the organism and improving immunity [44]. The A/G (albumin/globulin) ratio is the ratio of albumin to globulin, which can reflect the liver function and immune status of the organism, and a decrease in the white globule ratio indicates a decrease in albumin and an increase in the synthesis of globulin, which suggests that the organism has improved its immune function [45]. In this study, the serum levels of TP, ALB and GLB were elevated (*p* > 0.05) and the levels of A/G were reduced in fallow deer fed *Enterococcus faecalis*, which is consistent with the results of a previous study [46], suggesting that the protein metabolism of fallow deer can be improved to a certain extent by *Enterococcus faecalis*. ALT and AST are important aminotransferases in the body of the animal, reflecting the condition of the liver and playing a key role in the metabolism of TC and TG metabolism [47]. As shown in Figure 2, we also found that TC and TG in the serum of fallow deer fed *Enterococcus faecalis* increased significantly (*p* > 0.05), but there was no significant effect on both ALT and AST activities (*p* > 0.05), suggesting that *Enterococcus faecalis* promotes lipogenesis to a certain extent but has no adverse effect on liver function. In particular, at 65 d, *Enterococcus faecalis* significantly decreased LDH activity and elevated blood glucose levels (*p* < 0.05) and significantly increased phosphorus (P) levels (*p* < 0.05). These findings suggest that *Enterococcus faecalis* improves the synthesis and metabolism of proteins, sugars, and lipids in deer to a certain extent, thus contributing to the promotion of growth and development of the organism. The other biochemical parameters related to the parameters are in agreement with the values reported in the literature and are within the reference range [48].

Probiotics have been shown to have beneficial effects on the immune function of animals [49,50,51]. Previous studies reported that *Enterococcus faecalis* isolated from healthy chickens was able to increase serum IgY, as well as serum Newcastle disease virus haemagglutination inhibition (NDV HI) titres, the latter being a good indicator of signs of improvement in the intestinal immune response (intestinal-associated lymphoid tissues) suggesting that *Enterococcus faecalis* is able to stimulate the immune system of the animal [52]. A study on the effect of a combination of probiotics (*Enterococcus faecalis* and *Lactobacillus brevis*) on suckling piglets showed a significant increase in both serum IgA and IgG levels in the groups treated with the probiotics alone or in the mixture, whereas no significant increase in IgM was found [53]. In the present study, we found that IgA and IgM levels were significantly higher (*p* < 0.05) in the treated group than in the control group, increasing the level of immunity in weaned fallow deer. Similarly, Wang et al. reported that *Enterococcus faecalis* and *Clostridium butyricum* increased serum IgA and IgM levels in weaned piglets, respectively, and showed that the probiotics *Enterococcus faecalis* and Clostridium butyricum enhanced the immune function of the weaned piglet immune system [54]. In addition, inflammatory factors secreted by immune cells play a key role in the immune response, such as TNF-α, IL-6 and IL-8. Among them, TNF-α is a multifunctional cytokine involved in regulating the immune function and metabolic function of the body [55]. IL-6 can induce T cell activation, weight gain and differentiation and in appropriate amounts can enhance the body’s immune response and can also promote the production of IL-4 in the process of Th2 cell differentiation [56]. IL-8 plays an important role in preventing invasion of pathogenic microorganisms, the inflammatory response and promoting healing [57]. As can be seen in Figure 3, the levels of IL-6 and IL-8 in fallow deer increased significantly after 170 d of consuming feed supplemented with *Enterococcus faecalis*, which indicates that feeding *Enterococcus faecalis* can induce the organism to secrete IL-6 and IL-8 to enhance the immune function of the organism, thus reducing the occurrence of diseases.

To further explore the effects of *Enterococcus faecalis* on the gut microbiota of fallow deer, we characterised the *Enterococcus faecalis* of fallow deer at 35, 65 and 170 d post-treatment using high-throughput 16S rDNA sequencing. The α-diversity primarily assesses the diversity of the samples, reflecting the abundance and diversity of the microbial community [58]. Interestingly, we observed a significant increase difference (*p* < 0.05) in Chao1 and ACE indices only at day 35 of feeding diets containing *Enterococcus faecalis*, with no significant difference (*p* > 0.05) in α-diversity between the remaining control and treatment groups. A previous study found no significant changes in the α-diversity of the gut flora of forest musk deer after complex probiotic feeding [15]. It has also been shown that a comparison of feed supplementation with arginine on antler α-diversity showed a significant increase in ACE and Chao1 indices [59]. This suggests that faecal enterococci supplementation did not alter the abundance and diversity of the original microbial community of fallow deer. Regarding β-diversity, samples from the control and treatment groups were dispersed at 65 d and 170 d compared to 35 d (Figure 3), suggesting that differences between the gut microbiota gradually emerged as the feeding of *Enterococcus faecalis* proceeded. As shown in Figure 4A, we also found that microbes within the control group exhibited sparser associations with each other compared to the treatment group, suggesting that feeding *Enterococcus faecalis* resulted in closer coexistence between individuals in the microbiota. In a further development, the levels of *Firmicutes* and *Bacteroidetes*, the dominant faecal microorganisms, increased over time (Figure 4B). In other studies, *Firmicutes* and *Bacteroidetes* are the main drivers of polysaccharide fermentation and are also involved with the regulation of lipid and bile acid metabolism as well as energy homeostasis in the host [60,61]. Thus, *Firmicutes* and *Bacteroidetes* may have a synergistic symbiotic relationship, and an increase in their content may contribute to energy uptake/storage in the host and help the host gain weight. Within the phylum *Firmicutes*, *Ruminococcaceae* are present in the colonic mucosal biofilm of healthy individuals and are considered potentially beneficial bacteria because they actively regulate the intestinal environment and are associated with immune regulation and health homeostasis [62,63]. It was reported that *Lachnospiraceae*, another beneficial bacterium in the phylum *Firmicutes*, is thought to be symbiotic with *Ruminococcaceae* and plays a facilitating role in immune monitoring [64]. Importantly, at the genus level, our study showed an increase in the abundance of *Ruminocoecaceae* and *Lachnospiraceae* in the treatment group at 170 d (Figure 4C), confirming that *Enterococcus faecalis* increased the abundance of beneficial intestinal microbiota to some extent and exerted immune effects on the gut.

Previous evidence suggests that metabolites are an important interface between the gut microbiome and host health status [65]. From Figure 5A,B, we reveal clear differences in the metabolites of gut microbiota supplemented with *Enterococcus faecalis*. Comparing faecal metabolites at 35 d, we found that faecal metabolites were significantly altered in the treatment group at 170 d, with an increase in both up-regulated and down-regulated metabolites (Figure 5C). The significantly up-regulated 1-Oleoyl-L-α-lysophosphatidic acid at 35 d is one of the lipid mediators that regulate cell proliferation and differentiation through LAB receptors [66,67]. In contrast, the down-regulated Delta 8,14-Sterol is a precursor of trimethylolpropane oleate, which is broken down in vivo into components such as triglycerides, which can increase plasma concentrations of low-density lipoprotein cholesterol and very low-density lipoprotein cholesterol, thereby causing dyslipidaemia [68]. Importantly, the increase in the *Firmicutes* and *Bacteroidetes* is also consistent with the increased abundance of lipid metabolites observed in the metabolome. Therefore, the addition of *Enterococcus faecalis* to the feed of fallow deer promotes lipogenesis and inhibits lipid catabolism, which corroborates the results of weight gain. The metabolite 2-alpha-D-mannosyl-3-phosphoglycerate, increased at 170 days, has the ability to protect proteins and stabilise enzymes, and can act as an immunostimulant [69]. Studies have shown that *Enterococcus faecalis* can make the host use glycerol as one of the most important substrates in phospholipid biosynthesis and promote the production of 3-phosphoglycerol from endogenous glycerol, from which glycerol can be synthesised into glycerophospholipids, or glycerol can be used as a carbon source for the production of adenosine triphosphate (ATP) [70]. In addition, the increase in the anti-inflammatory and antioxidant Iberin-N-acetyl-cysteine [71] at 170 d also suggests that Enterococcus faecalis improved the intestinal flora environment. This is because Iberin-N-acetyl-cysteine prevents apoptosis and maintains long-term survival by activating the extracellular signal-regulated kinase pathway, which prevents apoptotic DNA fragmentation. Finally, KEGG showed that supplementation with *Enterococcus faecalis* significantly altered metabolites in fallow deer, including pathways of lipid metabolism, carbohydrate metabolism and phospholipid metabolism and suggests that intestinal flora may be related to these pathways. Therefore, we conclude that *Enterococcus faecalis* can improve the intestinal absorption of nutrients, which in turn can effectively increase the body weight of young animals, as well as regulate the intestinal microbiota and effectively prevent intestinal diseases in fallow deer.

## 5. Conclusions

In summary, the LAB strain *Enterococcus faecalis* can be used as a potential probiotic for deer. The results of this study showed that feed supplementation with *Enterococcus faecalis* improved body weight, growth performance and immune function of fallow deer and increased the aggregation of intestinal microorganisms in fallow deer. *Enterococcus faecalis* supplementation altered the community structure of fallow deer gut microbes and increased the number of beneficial bacteria. Specifically, at the phylum level, supplementation with *Enterococcus faecalis* increased the abundance of the bacteria phylum *Firmicutes*, and at the genus level, it increased the abundance of *Ruminocoecaceae* and *Lachnospiraceae*. Further, with metabolomic analyses it was revealed that supplementation with *Enterococcus faecalis* was able to alter significantly the metabolites of fallow deer, mainly modulating lipid metabolism, carbohydrate metabolism and phospholipid metabolism. To the best of our knowledge, this is the first report suggesting that the addition of LAB strain *Enterococcus faecalis* facilitates the growth and development of fallow deer, providing a basis for the future application of LAB strains as a probiotic in deer husbandry as well as for the reduction in intestinal diseases.

## Figures and Tables

**Figure 1 biology-13-00603-f001:**
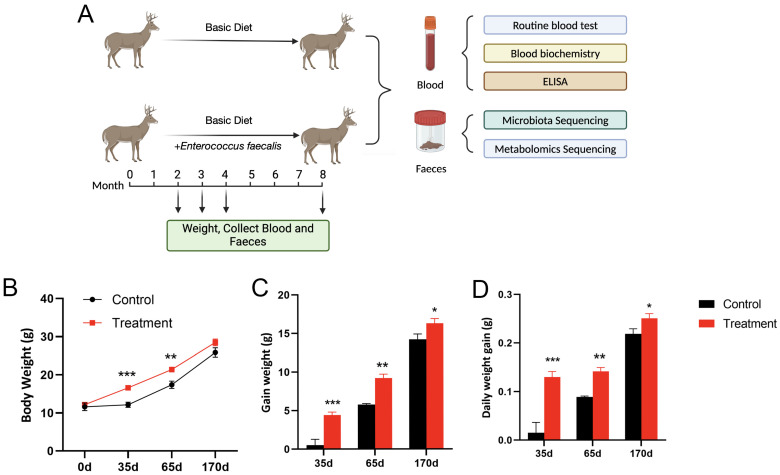
Experimental protocol and differences in body weights of fallow deer in the control and treatment groups. (**A**) Schematic diagram of the study protocol. Two-month-old fallow deer were fed a basal diet (control group) and a diet supplemented with *Enterococcus faecalis* (treatment group) for 170 d and were weighed, and blood and faecal samples were collected and analysed at 35 d, 65 d and 170 d. The (**B**) body weight curves, (**C**) body weight gain and (**D**) daily weight gain of control and treated fallow deer at 35 d, 65 d and 170 d were measured (see Appendix A). Results are expressed as mean ± SEM and significance is indicated by *p* value for comparisons with controls on the same day: * *p* < 0.05, ** *p* < 0.01 and *** *p* < 0.001 (*n* = 6 for all experiments).

**Figure 2 biology-13-00603-f002:**
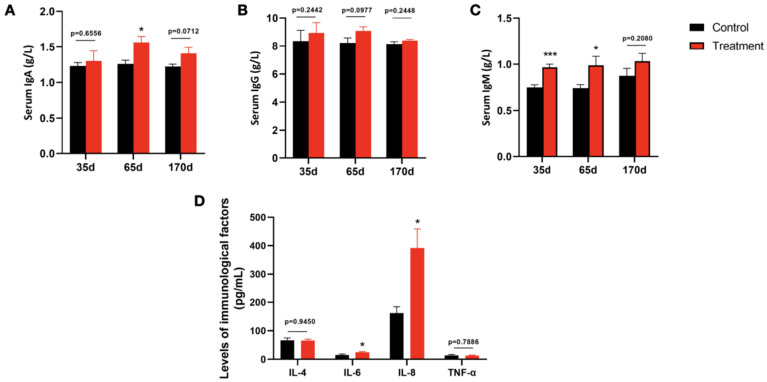
Effect of feed supplementation with *Enterococcus faecalis* on serum immune factors in fallow deer. Differences in (**A**) serum IgA, (**B**) serum IgM and (**C**) serum IgG between control and treated groups were assessed via blood biochemistry at 35 d, 65 d and 170 d (see Appendix A). (**D**) Differences in IL-4, IL-6, IL-8 and TNF-α between treated and control groups at 170 d were assessed via ELISA. Results are expressed as mean ± SEM and significance is indicated by *p* value for comparisons with controls on the same day: * *p* < 0.05 and *** *p* < 0.001 (*n* = 6 for all experiments).

**Figure 3 biology-13-00603-f003:**
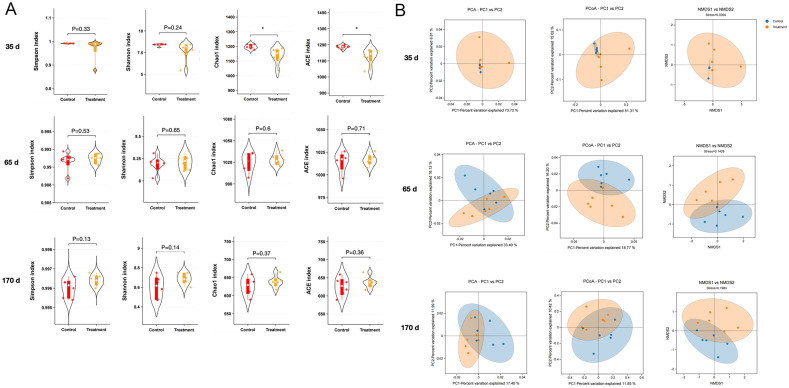
Changes in α- and β-diversity of gut microbiota in control and treatment groups. (**A**) Simpson’s index plot, Shannon’s index, Chao1 index, ACE index and rectangles represent box plots. (**B**) PCA, PCoA score plot and NMDS score plot based on binary_jaccard distance plot of gut microbial OTUs at35 d, 65 d and 170 d. The blue dots represent the control group, and the orange dots the represent treatment. Values are expressed as mean ± SEM and significance is indicated by *p* value for comparisons with controls on the same day: * *p* < 0.05.

**Figure 4 biology-13-00603-f004:**
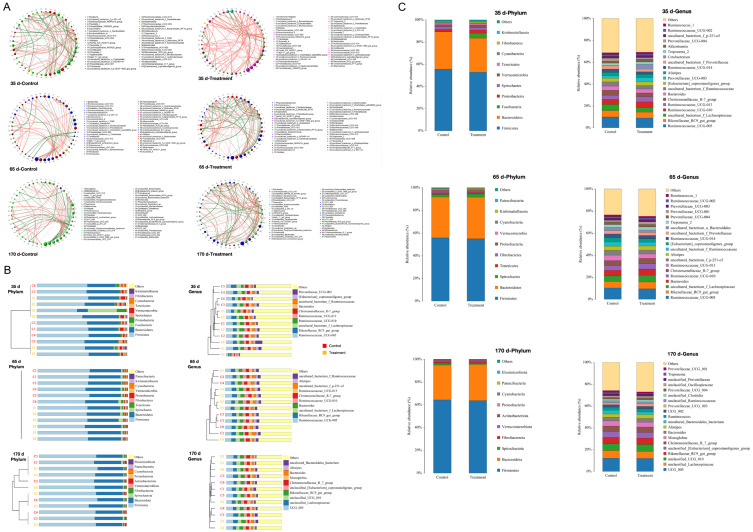
Effects of *Enterococcus faecalis* supplementation on the intestinal microbiota of fallow deer. (**A**) Microbial aggregation analyses of control and treatment groups at 35 d, 65 d and 170 d, and orange represents positive correlation and green represents negative correlation. (**B**) UPGMA analysis of arithmetic means (at the phylum level and genus level) for control and treatment groups at 35 d, 65 d and 170 d. (**C**) Relative abundance of the gut microbial community (at the phylum level and genus level) for control and treatment groups at 35 d, 65 d and 170 d.

**Figure 5 biology-13-00603-f005:**
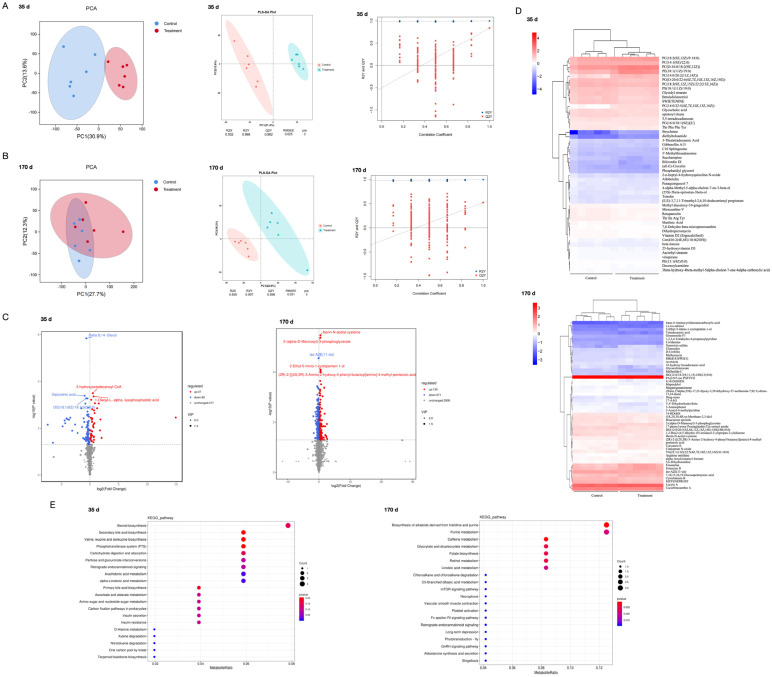
Metabolite composition analyses of control and treatment groups. (**A**) Validation plots of OPLS-DA model with PCA, PLS-DA and orthogonal partial least squares discriminant analysis of difference grouping for the control and treatment groups at 35 d. (**B**) Validation plots of the OPLS-DA model with PCA, PLS-DA and orthogonal partial least squares discriminant analysis of difference grouping for the control and treatment groups at 170 d. (**C**) Metabolite volcano plots of control and treated groups at 35 d and 170 d. *Enterococcus faecalis* after 35 d and 170 d of feed supplementation. (**D**) Heatmap made of the 50 metabolites with the most significant changes. (**E**) KEGG analysis plot.

**Table 1 biology-13-00603-t001:** Effect of *Enterococcus faecalis* on routine blood indices in fallow deer.

Construct	Group	Mean ± SEM			Test Effect	*p*-Value	η2
		35 d	65 d	170 d			
White blood cell (WBC) (10^9^/L)	Control	3.64 ± 0.36	3.04 ± 0.22	5.20 ± 0.62	Time	0.005	0.570
					Group	0.043	0.464
	Treatment	4.88 ± 0.47	4.25 ± 0.23 **	5.53 ± 0.41	Time × Group	0.443	0.105
Lymphocyte (Lymph) (10^9^/L)	Control	1.60 ± 0.07	1.26 ± 0.24	3.06 ± 0.41	Time	<0.001	0.810
					Group	0.184	0.237
	Treatment	2.18 ± 0.36	1.83 ± 0.13	3.40 ± 0.43	Time × Group	0.822	0.023
Monocyte (Mon) (10^9^/L)	Control	0.32 ± 0.05	0.32 ± 0.06	0.30 ± 0.04	Time	0.445	0.103
					Group	0.292	0.156
	Treatment	0.38 ± 0.05	0.40 ± 0.04	0.03 ± 0.04	Time × Group	0.348	0.137
Neutrophils (Gran) (10^9^/L)	Control	1.72 ± 0.28	1.48 ± 0.18	1.84 ± 0.19	Time	0.485	0.091
					Group	0.002	0.753
	Treatment	2.33 ± 0.17	2.07 ± 0.11 *	1.77 ± 0.06	Time × Group	0.270	0.172
Red blood cell (RBC) (10^12^/L)	Control	10.50 ± 0.46	10.48 ± 0.35	11.04 ± 0.08	Time	0.121	0.354
					Group	0.094	0.460
	Treatment	10.64 ± 0.04	11.13 ± 0.38	11.75 ± 0.49	Time × Group	0.667	0.072
Haemoglobin (HGB) (g/L)	Control	127.20 ± 6.70	127.4 ± 4.93	145.80 ± 1.16	Time	0.014	0.507
					Group	0.210	0.214
	Treatment	137.75 ± 4.03	133.25 ± 10.36	151.25 ± 3.12	Time × Group	0.813	0.021
Packed cell volume (PCV) %	Control	34.86 ± 2.08	34.54 ± 1.10	67.40 ± 1.00	Time	<0.001	0.985
					Group	0.007	0.725
	Treatment	36.70 ± 1.02 *	39.80 ± 1.19 *	70.13 ± 1.97	Time × Group	0.591	0.066
Platelet count (PLT) (10^9^/L)	Control	434.50 ± 3.93	271.75 ± 37.08	422.50 ± 15.87	Time	<0.001	0.741
					Group	0.053	0.490
	Treatment	459.00 ± 26.28	352.00 ± 62.58	603.25 ± 57.86 *	Time × Group	0.119	0.299
Plateletcrit (PCT) %	Control	0.17 ± 0.01	0.12 ± 0.01	0.27 ± 0.01	Time	<0.001	0.887
					Group	0.040	0.523
	Treatment	0.19 ± 0.01	0.16 ± 0.03	0.40 ± 0.05 *	Time × Group	4.064	0.404

Values are expressed as mean ± SEM, with different * in the rows indicating a significant difference compared to the control group on the same day; * for *p* < 0.05 and ** for *p* < 0.01.

**Table 2 biology-13-00603-t002:** Effect of *Enterococcus faecalis* on blood biochemical indices of fallow deer.

Construct	Group	Mean ± SEM			Test Effect	*p*-Value	η2	Detection Range
		35 d	65 d	170 d				
Total protein (TP) (g/L)	Control	48.06 ± 1.32	51.47 ± 0.98	50.60 ± 2.10	Time	0.065	0.269	<100
					Group	0.075	0.310	
	Treatment	51.22 ± 1.13	51.63 ± 1.93	55.81 ± 1.21	Time × Group	0.215	0.157	
Albumin (ALB) (g/L)	Control	26.39 ± 0.49	32.58 ± 0.26	32.49 ± 1.64	Time	<0.001	0.720	<60
					Group	0.192	0.203	
	Treatment	28.15 ± 0.70	31.94 ± 1.56	35.08 ± 0.92	Time × Group	0.314	0.134	
Globulin (GLB) (g/L)	Control	20.45 ± 1.50	18.71 ± 0.66	18.39 ± 0.73	Time	0.072	0.243	<40
					Group	0.085	0.268	
	Treatment	23.07 ± 1.19	19.35 ± 1.59	20.64 ± 0.96	Time × Group	0.635	0.04	
Albumin/Globulin (A/G)	Control	1.22 ± 0.05	1.74 ± 0.05	1.82 ± 0.09	Time	<0.001	0.653	
					Group	0.789	0.008	
	Treatment	1.24 ± 0.08	1.74 ± 0.18	1.72 ± 0.09	Time × Group	0.748	0.021	
Total Cholesterol (TC) (mmol/L)	Control	1.57 ± 0.23	1.49 ± 0.03	1.65 ± 0.11	Time	0.006	0.490	<5.17
					Group	0.439	0.068	
	Treatment	1.47 ± 0.05	1.50 ± 0.06	1.86 ± 0.08	Time × Group	0.126	0.219	
Triglycerides (TG) (mmol/L)	Control	0.24 ± 0.04	0.16 ± 0.02	0.14 ± 0.01	Time	0.018	0.396	<1.71
					Group	0.006	0.594	
	Treatment	0.27 ± 0.02	0.26 ± 0.02 **	0.20 ± 0.02 *	Time × Group	0.496	0.069	
Aspartate Amino Acid Transferase (AST) (U/L)	Control	134.19 ± 14.78	109.38 ± 7.71	89.35 ± 10.19	Time	0.023	0.332	<140
					Group	0.010	0.504	
	Treatment	103.33 ± 3.98	90.06 ± 6.20	87.50 ± 5.53	Time × Group	0.338	0.101	
Alanine aminotransferase (ALT) (U/L)	Control	66.42 ± 5.50	84.09 ± 4.26	66.76 ± 6.38	Time	0.034	0.288	<140
					Group	0.194	0.163	
	Treatment	60.32 ± 2.17	70.63 ± 6.62	69.24 ± 3.61	Time × Group	0.303	0.112	
Glucose (GLU) (mmol/L)	Control	7.27 ± 1.04	4.97 ± 0.77	9.38 ± 0.80	Time	0.014	0.437	5.89–9.41
					Group	0.901	0.002	
	Treatment	6.19 ± 0.69	7.31 ± 0.46 *	8.34 ± 0.18	Time × Group	0.082	0.269	
Lactate dehydrogenase (LDH) (U/L)	Control	695.71 ± 67.92	643.91 ± 24.96	628.83 ± 43.80	Time	0.399	0.088	436–680
					Group	0.001	0.652	
	Treatment	557.49 ± 14.80	500.02 ± 24.78 **	601.82 ± 25.16	Time × Group	0.304	0.112	
Phosphorus (P) (mmol/L)	Control	2.64 ± 0.10	2.90 ± 0.09	2.63 ± 0.12	Time	0.027	0.308	2.46–4.16
					Group	<0.001	0.791	
	Treatment	3.51 ± 0.12 ***	3.36 ± 0.13 *	3.03 ± 0.92 *	Time × Group	0.087	0.218	

Values are expressed as mean ± SEM, with different * in the rows indicating a significant difference compared to the control group; * for *p* < 0.05, ** for *p* < 0.01 and *** for *p* < 0.001.

## Data Availability

All the data involved in this experiment are available through the corresponding author.

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
