# Peer review of "Effects of Probiotic Supplementation on Body Weight, Growth Performance, Immune Function, Intestinal Microbiota and Metabolites in Fallow Deer"

_biology, 2024, doi:10.3390/biology13080603_

Round 1

Reviewer 1 Report

Comments and Suggestions for Authors

1. Line 204. higher by 23.07% (p < 0.01), 59.97% (p < 0.001), and 55.56% (p < 0.001) - it should be replaced by 0,01.

2. Figure 3, 4, 5 are too small. It is difficult to read the axes labels and words on the Figures.

3. On my opinion it is not correct to compare the results with different statistical significance on 400-408 lines.

Author Response

Answer to the Reviewers` Comments

Dear reviewers of Biology:

We are very glad to receive your mail. Thank you for your letter and for your comments on our manuscript entitled "Effects of probiotic supplementation on body weight, immune function, intestinal microbiota and metabolites in fallow deer". Those comments are valuable and very helpful for revising and improving our paper. Based on your comment and request, we have made extensive modification on the original manuscript. Detailed modifications are attached below.

Finally, we acknowledge the reviewer`s comments and suggestions very much, which are valuable in improving the quality of our manuscript.

Corresponding authors:

Name: Qingyun Guo

  • mail: guoqingyun1987@163.com

Comments 1: Line 204. higher by 23.07% (p < 0.01), 59.97% (p < 0.001), and 55.56% (p < 0.001) - it should be replaced by 0,01.

Response 1: We totally understand the reviewer`s concern. We have changed “59.97% (p < 0.001), and 55.56% (p < 0.001)” to “59.97% (p < 0.01), and 55.56% (p < 0.01)” in the article. and re-upload the image.

Pleases see Result 3.1 line 226.

Comments 2: Figure 3, 4, 5 are too small. It is difficult to read the axes labels and words on the Figures.

Response 2: We feel sorry for the inconvenience brought to the reviewer. We have modified the resolution of Figure 3, 4, 5 and enlarged the image and reuploaded the image in the manuscript.

Pleases see Figure 3, 4, 5.

Comments 3: On my opinion it is not correct to compare the results with different statistical significance on 400-408 lines.

Response 3: Thank you for your rigorous nice suggestion. In order to correctly express the meaning of the result, we have changed “In this study, the addition of Enterococcus faecalis to the diet significantly increased body weight, total weight gain and average daily gain of fallow deer after 35 d of feeding compared to the control group, indicating that the addition of Enterococcus faecalis to the diet significantly promoted the growth and development of weaned fallow deer (p < 0.001). Similarly, at 65 d, these indicators reflecting the growth performance of fallow deer were significantly improved (p < 0.01).” to “In this study, the addition of Enterococcus faecalis to the diet significantly increased body weight, total weight gain and average daily gain of fallow deer after 35 d of feeding compared to the control group (p < 0.001), indicating that dietary addition of Enterococcus faecalis may improve the digestion of nutrients in fallow deer. Similarly, at 65 d, these indicators reflecting weight gain of fallow deer were significantly improved (p < 0.01).” in the manuscript.

Pleases see Discussion section line 441-446.

Reviewer 2 Report

Comments and Suggestions for Authors

In this study, the effect of Enterococcus faecalis as probiotic feed additive on the growth performance, immunity, metabolism and microbiome of deer was evaluated. Recently, there is an urgent need for such research to minimize the use of antibiotics and its consequences on human and animal health. However, the main concern in this study is about how the dose of the probiotic was selected. Mentioning the ratio of the product of probiotic used in the diet is not enough. Information on the number of CFU/g diet is very important. To get probiotic effect the number of the probiotic live cells should not be less than 107 per gram diet. This point should be clearly shown. For statistical analyses, I belive that two way anova for repeated measurments are more suitable for biochemical paramters in serum and othe reapeated mesurments.

Also for the diet, information on the ingredients of the concentrate diet and its chemical analysis should be shown. Also, it is not clear if the concentrate diet, carrot, alfa alfa, and other ingredients were offered as total mixture ration or separately.

Specific comments:

84 : such as Lactobacillus plantarum [24], ..plantarum should be italic

115 : Throughout the experimental period, fallow deer were fed once a day at 9:00 and 15:00 and were self-consumed,……do you mean twice a day??? Please, revise this sentence.

116:  with each deer consuming approx- imately 500 g of feed per day….It is not logic to feed animals the same quantity of feed for 170 days??..the amount should be changed to meet the nutritional requirments of growth with time?????

Section 2.2. need reference and more details on the procedure used for fecal samples collection

131-134: Blood samples were collected and placed in collection tubes con-  taining anticoagulant, the anticoagulant tubes were gently shaken upside down and centrifuged in a centrifuge (Eppendorf,Germany) at 3000 rpm for 10 min and the supernatant was taken as plasma. After that, we measured the plasma levels of blood cells by BC-2800Vet Haematology Analyzer (Mindray, China)…Revise this paragraph, hematological paramters should be done for a whole sample blood before plasma separation.

134: we measured the plasma levels of blood cells by ….remove cells

146: im- munoglobulin (IgA, IgM, IgG) levels…Please, make sure that these parameters can be analysed by BS-420 fully automated biochemistry..and for biochemical parameters mention the source of the kit used and its sensitivity.

150: ELISA kits (Cloud-Clone Corp., China) were used to detect the levels of the immunological factors TNF-α, IL-4, IL-6 and IL-8 in serum… Are these kits specific for deer cytokines..Please, mention the cross reactivity, inter and intra assay CV.

For growth performance results: there was not any description for the procedure of evaluating growth parameters in MM section

For the statistical analysis: the model used for analysis should be shown. In this study there is a need for using more than one model, such as model for repeated factorial measurements and another for one observed data. Also, a full description for the statistical progrmas and analyses procedures/models for microbiome and metabolomic analyses should be shown.

The resolution of figures, specially, figures 3 and 4 need more attaention. The quality and the font of the figures are not good enough.

 509 : Therefore, the addition of Enterococcus faecalis to the feed of fallow deer promotes lipogen-esis and inhibits lipid metabolism,..Do you mean inhibits lipid catabolism???

References: 86 references are too much. I suggest using only related research to rumiants  and removing those for non-ruminant animals such as pigs, rabbits, and poultry

For more specific articles, you may need such article:Mousa GA, Allak MA, Shehata MG, Hashem NM, Hassan OGA. Dietary Supplementation with a Combination of Fibrolytic Enzymes and Probiotics Improves Digestibility, Growth Performance, Blood Metabolites, and Economics of Fattening Lambs. Animals (Basel). 2022 Feb 15;12(4):476. doi: 10.3390/ani12040476. PMID: 35203183; PMCID: PMC8868385.

Comments on the Quality of English Language

Minor revision

Author Response

Answer to the Reviewers` Comments

Dear reviewers of Biology:

We are very glad to receive your mail. Thank you for your letter and for your comments on our manuscript entitled "Effects of probiotic supplementation on body weight, immune function, intestinal microbiota and metabolites in fallow deer". Those comments are valuable and very helpful for revising and improving our paper. Based on your comment and request, we have made extensive modification on the original manuscript. Detailed modifications are attached below.

Finally, we acknowledge the reviewer`s comments and suggestions very much, which are valuable in improving the quality of our manuscript.

Corresponding authors:

Name: Qingyun Guo

  • mail: guoqingyun1987@163.com

Comments 1: In this study, the effect of Enterococcus faecalis as probiotic feed additive on the growth performance, immunity, metabolism and microbiome of deer was evaluated. Recently, there is an urgent need for such research to minimize the use of antibiotics and its consequences on human and animal health. However, the main concern in this study is about how the dose of the probiotic was selected. Mentioning the ratio of the product of probiotic used in the diet is not enough. Information on the number of CFU/g diet is very important. To get probiotic effect the number of the probiotic live cells should not be less than 107 per gram diet. This point should be clearly shown.

Response 1: We gratefully appreciate for your valuable suggestion. The concentration of Enterococcus faecalis added to the diets we fed to fallow deer was ≥2.0×1010CFU/g. We have supplemented this in section 2.1 Animal Management of Material Methods.

Pleases see the Materials and Methods section in line 120.

Comments 2: For statistical analyses, I belive that two way anova for repeated measurments are more suitable for biochemical paramters in serum and othe reapeated mesurments.

Response 2: Thank you for your rigorous nice suggestion. We have added to 2.7 Statistical Analyses in the Materials and Methods section the supplement “Analysis of variance (ANOVA) approaches were used to evaluate Serum biochemical parameters.”

Pleases see the 2.7 Statistical Analyses section in line 214.

Comments 3: Also for the diet, information on the ingredients of the concentrate diet and its chemical analysis should be shown. Also, it is not clear if the concentrate diet, carrot, alfa alfa, and other ingredients were offered as total mixture ration or separately.

Response 3: We totally understand the reviewer`s concern. The concentrate feed mentioned in our manuscript is a deer concentrate feed (Junwei, Tianjin, China) with a feed formulation of 50% maize, 26% soya bean meal, 11% bran, 10% barley, 2% calcium phosphate, 1% salt. In addition, the concentrate feeds, carrots, alfalfa, and other ingredients were provided as total mixed diets. All the above modifications were added in the “2.1 Animal Management” section of Materials and Methods.

Pleases see the 2.1 Animal Management section in line 122.

Comments 4: 84 : such as Lactobacillus plantarum [24], ..plantarum should be italic

Response 4: Thank you so much for your careful check. We've checked the full text for bacteria involved and changed them all to italics.

Pleases see the Introduction section in line 88.

Comments 5: 115 : Throughout the experimental period, fallow deer were fed once a day at 9:00 and 15:00 and were self-consumed,……do you mean twice a day??? Please, revise this sentence.

Response 5: Thank you for your rigorous nice suggestion. We have changed “once a day” to “twice a day” in line 123.

Pleases see the 2.1 Animal Management section in line 123.

Comments 6: 116: with each deer consuming approximately 500 g of feed per day….It is not logic to feed animals the same quantity of feed for 170 days??..the amount should be changed to meet the nutritional requirments of growth with time?????

Response 6: We gratefully appreciate for your valuable suggestion. We add to “Throughout the experimental period, fallow deer were fed twice a day at 9:00 and 15:00 and were self-consumed, with each deer consuming approximately 500 g of feed per day.” after “The quantity of feed should be varied over time to meet the nutritional requirements for growth”.

Pleases see the 2.1 Animal Management section in line 126.

Comments 7: Section 2.2. need reference and more details on the procedure used for fecal samples collection

Response 7: Thank you for your rigorous nice suggestion. Section 2.2. cites Zhang et al.'s sample collection procedure for fallow deer faeces to add more detail to the discriminating sample collection procedure.

Pleases see the 2.2 Faecal Collection and Handling section.

Comments 8: 131-134: Blood samples were collected and placed in collection tubes containing anticoagulant, the anticoagulant tubes were gently shaken upside down and centrifuged in a centrifuge (Eppendorf,Germany) at 3000 rpm for 10 min and the supernatant was taken as plasma. After that, we measured the plasma levels of blood cells by BC-2800Vet Haematology Analyzer (Mindray, China)…Revise this paragraph, hematological paramters should be done for a whole sample blood before plasma separation.

Response 8: Thank you so much for your careful check. We tested haematological parameters using untreated fallow deer whole blood. We have changed this part to “Blood samples were collected and placed in collection tubes containing anticoagulant, the anticoagulant tubes were gently shaken upside down. After that, we measured the plasma levels of blood cells by BC-2800Vet Haematology Analyzer (Mindray, China).”

Pleases see the 2.3 Blood Collection and Processing section.

Comments 9: 134: we measured the plasma levels of blood cells by ….remove cells

Response 9: Thank you so much for your careful check. We have deleted the word “cells”.

Pleases see the 2.3 Blood Collection and Processing section.

Comments 10: 146: immunoglobulin (IgA, IgM, IgG) levels…Please, make sure that these parameters can be analysed by BS-420 fully automated biochemistry..and for biochemical parameters mention the source of the kit used and its sensitivity.

Response 10: Thank you for pointing out this issue. We use the Deer Immunoglobulin IgG / A / M Assay Kit (Meimian, Jiangsu, China). We measure immunoglobulin (IgA, IgM, IgG) levels. And “Serum is analysed biochemically by biochemical marker kits (Zhongsheng Beizhong Biotechnology Co., Ltd., Beijing, China)”. The remaining biochemical parameters are complemented by the sensitivity of all biochemical parameters in Table 2.

Pleases see the 2.3 Blood Collection and Processing section and Table 2.

Comments 11: 150: ELISA kits (Cloud-Clone Corp., China) were used to detect the levels of the immunological factors TNF-α, IL-4, IL-6 and IL-8 in serum… Are these kits specific for deer cytokines..Please, mention the cross reactivity, inter and intra assay CV.

Response 11: We gratefully appreciate for your valuable suggestion. The applicable animal of the ELISA kit used in this study is bovine, but the homology between deer and bovine is high. We also describe the kit's cross-reactivity, inter- and intra-assay CVs. Specifically as follows: “The substances detected in the TNF-α, IL-4, IL-6 and IL-8 kits showed no significant cross-reactivity with other similar substances and The intra assay CV was less than 10% and the inter assay CV was less than 12%.”

Pleases see the 2.4 Enzyme Linked Immunosorbent Assay (ELISA) section.

Comments 12: For growth performance results: there was not any description for the procedure of evaluating growth parameters in MM section

Response 12: We feel sorry for the distress brought to the reviewer. The procedure for evaluating growth performance is not covered in the manuscript, so we have deleted the part of the text that refers to “growth performance”.

Pleases see the full text.

Comments 13: For the statistical analysis: the model used for analysis should be shown. In this study there is a need for using more than one model, such as model for repeated factorial measurements and another for one observed data. Also, a full description for the statistical progrmas and analyses procedures/models for microbiome and metabolomic analyses should be shown.

Response 13: We gratefully appreciate for your valuable suggestion. We supplemented the modeling of the statistical analyses in “2.5 Fecal DNA Extraction, PCR Amplification and Sequencing of Intestinal Microbiota”, “2.6 Non-targeted metabolomics analysis of Intestinal Microbiota metabolites” and “2.7 Data Analysis”. We have added“Using the Usearch software [?] Clustering of Reads at 97.0% similarity level, obtaining OTUs. using the SILVA database (v.123) the classifications were assigned to OTUs with a confidence level of 70% for the RDP classifiers [?] . Alpha diversity indices including ACE, Chao1, Shannon and Simpson indices were calculated using QIIME2 software (v.2020.8). Statistical significance between groups was determined by one-way analysis of variance (ANOVA). The structure of microbial communities among samples was analyzed by Beta diversity analysis using binary_Jaccard distance metric and visualized by principal coordinate analysis (PCoA), PCoA, NMDS, and UPGMA.” and “Experiments were performed in at least five independent biological replicates and at least two independent technical replicates.” to the Materials and Methods section.

Pleases see the Materials and methods 2.5, 2.6 and 2.7 section.

Comments 14: The resolution of figures, specially, figures 3 and 4 need more attaention. The quality and the font of the figures are not good enough.

Response 14: We feel sorry for the inconvenience brought to the reviewer. We have increased the resolution of Figures 3, 4, and 5 in the article and enlarged them by reinserting them in the manuscript.

Pleases see Figure 3, 4, 5.

Comments 15: 509 : Therefore, the addition of Enterococcus faecalis to the feed of fallow deer promotes lipogen-esis and inhibits lipid metabolism,..Do you mean inhibits lipid catabolism???

Response 15: Thank you so much for your careful check. We've changed “inhibit lipid metabolism” to “inhibit lipid catabolism” in the article.

Pleases see the Discussion section in line 562.

Comments 16: References: 86 references are too much. I suggest using only related research to rumiants  and removing those for non-ruminant animals such as pigs, rabbits, and poultry

Response 16: Thank you for pointing out this issue. We removed studies related to non-ruminants such as pigs, rabbits and poultry, reducing the number of references from 86 to 69.

Pleases see the References section.

Reviewer 3 Report

Comments and Suggestions for Authors

The manuscript submitted by Wang et al. describes the effect of probiotic supplementation on body weight, growth performance, immune function, intestinal microbiota, and metabolites in fallow deer. The manuscript is very interesting and well-written and I recommend its publication. I have a few comments that might be considered.

Graphic abstract, Family names in the box should be written in italic

Please update the introduction with more recent citation

Please check all materials used in this study and provide the supplier, City and country for each

Line 235: Ebterococcus fecalis should be in italic

Figure 5. Not clear, and I could not read any of these metabolites. Maybe authors can provide a higer resolution figure

Line 398: Bacillus licheniformis and Saccharomyces cerevisiae should be in italic

Discussion. I suggest to discuss some of the modulated metabolites.

References: Please write bacterial species in italic in all references

Additional minor comments, please see attached file

Comments on the Quality of English Language

Author Response

Answer to the Reviewers` Comments

Dear reviewers of Biology:

We are very glad to receive your mail. Thank you for your letter and for your comments on our manuscript entitled "Effects of probiotic supplementation on body weight, immune function, intestinal microbiota and metabolites in fallow deer". Those comments are valuable and very helpful for revising and improving our paper. Based on your comment and request, we have made extensive modification on the original manuscript. Detailed modifications are attached below.

Finally, we acknowledge the reviewer`s comments and suggestions very much, which are valuable in improving the quality of our manuscript.

Corresponding authors:

Name: Qingyun Guo

  • mail: guoqingyun1987@163.com

Comments 1: Graphic abstract, Family names in the box should be written in italic

Response 1: Thank you so much for your careful check. We have changed Ruminococcaceae and Lacertococcaceae to italics in the graphical abstract and uploaded the revised images in the manuscript.

Pleases see the Graphic abstract in line 45.

Comments 2: Please update the introduction with more recent citation

Response 2: We learnt about the latest article by Shehata et al. on restoring healthy gut microbiota in animals using plant-derived substances as an alternative to feed additives, which is supplemented in the manuscript.

Pleases see the References 14.

Comments 3: Please check all materials used in this study and provide the supplier, City and country for each

Response 3: Thank you for pointing out this issue. We have checked the materials and equipment covered in the Material Methods section of the article and provided the supplier, city and country for each. For example, we changed (Eppendorf, Germany) to (Eppendorf, Hamburg, Germany) in the material methods section 2.3.

Pleases see the Materials and Methods section.

Comments 4: Line 235: Enterococcus fecalis should be in italic

Response 4: Thank you so much for your careful check. We have changed the formatting of the article Enterococcus fecalis to italics.

Pleases see line 263.

Comments 5: Figure 5. Not clear, and I could not read any of these metabolites. Maybe authors can provide a higer resolution figure

Response 5: We feel sorry for the inconvenience brought to the reviewer. We have modified the resolution of Figure 5 and enlarged the image and changed the image in the manuscript.

Pleases see Figure 5.

Comments 6: Line 398: Bacillus licheniformis and Saccharomyces cerevisiae should be in italic

Response 6: Thank you so much for your careful check. We have changed the formatting of the article Bacillus licheniformis and Saccharomyces cerevisiae to italics.

Pleases see line 439.

Comments 7: Discussion. I suggest to discuss some of the modulated metabolites.

Response 7: We gratefully appreciate for your valuable suggestion. We provide further discussion of regulatory metabolites and add to the discussion section of the manuscript as follows: “Previous evidence suggests that metabolites are an important interface between the gut microbiome and host health status [63]. From Figure 5A, B, we reveal clear differences in the metabolites of gut microbes supplemented with Enterococcus faecalis. Comparing faecal metabolites at 35 d, we found that faecal metabolites were significantly altered in the treatment group at 170 d, with an increase in both up-regulated and down-regulated metabolites (Figure 5 C). The significantly up-regulated 1-Oleoyl-L-α-lysophosphatidic acid at 35 d is one of the lipid mediators that regulate cell proliferation and differentiation through LAB receptors [64,65]. In contrast, the down-regulated Delta 8,14-Sterol is a precursor of trimethylolpropane oleate, which is broken down in vivo into components such as triglycerides, which can increase plasma concentrations of low density lipoprotein cholesterol and very low-density lipoprotein cholesterol, thereby causing dyslipidaemia [66]. Importantly, the increase in the Firmicutes and Bacteroidetes is also consistent with the increased abundance of lipid metabolites observed in the metabolome. Therefore, the addition of Enterococcus faecalis to the feed of fallow deer promotes lipogenesis and inhibits lipid catabolism, which corroborates the results of weight gain. The metabolite 2-alpha-D-mannosyl-3-phosphoglycerate, increased at 170 days, has the ability to protect proteins and stabilise enzymes, and can act as an immunostimulant [67]. Studies have shown that Enterococcus faecalis can make the host use glycerol as one of the most important substrates in phospholipid biosynthesis, and promote the production of 3-phosphoglycerol from endogenous glycerol, from which glycerol can be synthesised into glycerophospholipids, or the use of glycerol as a carbon source for the production of adenosine triphosphate (ATP) [68]. In addition, the increase in the anti-inflammatory and antioxidant Iberin-N-acetyl-cysteine [69] on 170 d also suggests that Enterococcus faecalis improved the intestinal flora environment. This is because Iberin-N-acetyl-cysteine prevents apoptosis and maintains long-term survival by activating the extracellular signal-regulated kinase pathway, which prevents apoptotic DNA fragmentation. Finally, KEGG showed that supplementation with Enterococcus faecalis significantly altered metabolites in fallow deer, including pathways of lipid metabolism, carbohydrate metabolism, and phospholipid metabolism, and suggests that intestinal flora may be related to these pathways. Therefore, we conclude that Enterococcus faecalis can improve the intestinal absorption of nutrients, which in turn can effectively increase the body weight of young animals, as well as regulate the intestinal microbiota and effectively prevent intestinal diseases in fallow deer.”

Pleases see the Discussion section.

Comments 8: References: Please write bacterial species in italic in all references

Response 8: Thank you so much for your careful check. We checked all references that dealt with bacteria and wrote the bacterial species in italics.

Pleases see the References section.

Comments 9: Additional minor comments, please see attached file

Response 9: We gratefully thanks for the precious time the reviewer spent making constructive remarks. We studied the recent article by Shehata et al. to understand that because metabolomics involves events occurring downstream of gene expression it can therefore reflect phenotype more directly, and added to the manuscript.

Pleases see the Reference 63.

Round 2

Reviewer 2 Report

Comments and Suggestions for Authors

The study was modulated according to some of the comments addressed in the previous revision. However, the main concerns regarding the number of live probiotic per each gram diet is still not clear. How many live bacteria will be available per gram diet with 0.002%  of a product contains 2.0×1010CFU/g. The second concern regarding growth performance, deleting the term growth performance is not what I mean, but how and when body weight of animals were recorded and and how the weight gain and daily weight gain were calculated. The sentence added to show that the feed quantity was changed with time to meet the growth of deer should be written by a scientific way (sentence re-editing with proper English and on which base the quantity was changed, animals weight or feed consumption, and the composition of the concentrate  diet should be shown. Lastly,   the data of repeated data should be re analyzed by mixed factorial ANOVA..  

Comments on the Quality of English Language

The quality of English is sufficient but needs minor revision

Author Response

Answer to the Reviewers` Comments

We are pleased to hear from you again. Thank you very much for your comments for the manuscript entitled "Effects of probiotic supplementation on body weight, growth performance, immune function, intestinal microbiota and metabolites in fallow deer". Based on your comment and request, we have made detailed modification on the original manuscript. The modifications are attached below. We have revised the English quality of the article accordingly.

Finally, we acknowledge the reviewer`s comments and suggestions very much, which are valuable in improving the quality of our manuscript.

Corresponding authors:

Name: Qingyun Guo

E-mail: guoqingyun1987@126.com

Comments 1: The main concerns regarding the number of live probiotic per each gram diet is still not clear. How many live bacteria will be available per gram diet with 0.002% of a product contains ≥2.0×1010 CFU/g.

Response 1: Thank you for your rigorous nice suggestion. 0.002% faecal enterococci in the diet is approximately 0.01 g. Therefore, Enterococcus faecalis per gram of diet is approximately 2.0×108 CFU/g.

Pleases see 2.1 Animal Management in line 123.

Comments 2: The concern regarding growth performance, deleting the term growth performance is not what I mean, but how and when body weight of animals were recorded and and how the weight gain and daily weight gain were calculated.

Response 2: We apologize for misinterpreting the reviewers' comments. We have reintroduced the term “growth performance” in the manuscript, and the “2.2 Weight determination” section of the Materials and Methods describes in detail how and when to record the body weight of an animal, and how to calculate gain weight and daily weight gain. Details are as follows: At 0 d, fallow deer were weighed 2 h before morning feeding as initial weight. On 35 d, 65 d, and 170 d, all animals were weighed 2 h before morning feeding, and growth curves for fallow deer were made using days as the horizontal axis and body weight as the vertical axis. Gain weight was obtained by subtracting the initial weight from the final body weight. Daily weight gain is the gain weight divided by the number of days in the study.

Pleases see 2.2 Weight determination.

Comments 3: The sentence added to show that the feed quantity was changed with time to meet the growth of deer should be written by a scientific way (sentence re-editing with proper English and on which base the quantity was changed, animals weight or feed consumption, and the composition of the concentrate diet should be shown.

Response 3: We gratefully appreciate for your valuable suggestion. We have re-edited the sentences in correct English and added the basis for changing the amount of feed, the weight of the animal or feed consumption and the composition of the concentrate feed. The details are as follows: “Specifically speaking, the treatment group had a mixed diet of 22% concentrate (Junwei, Tianjin, China), 26% carrots, 27% dry alfalfa grass, 25% silage and 0.002% Enterococcus faecalis (≥2.0×108CFU/g). The deer concentrate feed (Junwei, Tianjin, China) with a feed formulation of 50% maize, 26% soya bean meal, 11% bran, 10% barley, 2% calcium phosphate, 1% salt. The mixed diet covers the recommended nutritional requirements [29 ,30]. The diet was adjusted daily based on the previous day's leftovers to ensure that approximately 1% of the feed was left in the trough at the second day feeding.”

Pleases see 2.1 Animal Management in line 123-128.

Comments 4: The data of repeated data should be re analyzed by mixed factorial ANOVA.

Response 4: We totally understand the reviewer`s concern. According to Zhen et al. (Zhen J, Ren Y, Zhang H, Yuan X, Wang L, Shen H, Liu P, Chen Y. Effect of Different Dietary Regimes on the Gut Microbiota and Fecal Metabolites of Père David's Deer. animals (Basel). 2022; 12(5): 584. doi: 10.3390/ani12050584. PMID: 35268151; PMCID: PMC8909101.) of statistical analyses, the significance of the present experiment was primarily tested using the Student's t-test between the two sets of experimental values. For example, in Fig. 1C, at 65 days was used to determine whether the addition of Enterococcus faecalis to the diet after 65 days of feeding had a significant enhancing effect on the growth performance of fallow deer by comparing the body weight gain of the control group with that of the treatment group.

Pleases see “2.8 Statistical Analyses.

Comments 5: The quality of English is sufficient but needs minor revision.

Response 5: We have made detailed corrections to both multiple grammatical errors and inaccuracies in expression throughout the text. In addition, improper formatting and bacterial italics have been addressed. For example, we change “Enterococcus faecalis is a parthenogenetic anaerobic Gram-positive lactic acid bacterium and an intrinsic bacterium of the gastrointestinal tract of humans and animals [21].” to “Enterococcus faecalis is a parthenogenetic anaerobic gram-positive LAB and an inherent bacteria of the gastrointestinal tract of humans and animals [21].” in line 91-92.

Pleases see the full text.

Reviewer 3 Report

Comments and Suggestions for Authors

Accept

Comments on the Quality of English Language

Accept

Author Response

Answer to the Reviewers` Comments

We are pleased to hear from you again. Thank you very much for your comments for the manuscript entitled "Effects of probiotic supplementation on body weight, growth performance, immune function, intestinal microbiota and metabolites in fallow deer". Based on your comment and request, we have made detailed modification on the original manuscript. The modifications are attached below. We have revised the English quality of the article accordingly.

Finally, we acknowledge the reviewer`s comments and suggestions very much, which are valuable in improving the quality of our manuscript.

Corresponding authors:

Name: Qingyun Guo

E-mail: guoqingyun1987@126.com

Comments 1: Minor editing of English language required.

Response 1: We have made detailed corrections to both multiple grammatical errors and inaccuracies in expression throughout the text. In addition, improper formatting and bacterial italics have been addressed. For example, we change “Enterococcus faecalis is a parthenogenetic anaerobic Gram-positive lactic acid bacterium and an intrinsic bacterium of the gastrointestinal tract of humans and animals [21].” to “Enterococcus faecalis is a parthenogenetic anaerobic gram-positive LAB and an inherent bacteria of the gastrointestinal tract of humans and animals [21].” in line 91-92.

Pleases see the full text.
